# Involvement of Substance P (SP) and Its Related NK1 Receptor in Primary Sjögren’s Syndrome (pSS) Pathogenesis

**DOI:** 10.3390/cells12101347

**Published:** 2023-05-09

**Authors:** Pamela Rosso, Elena Fico, Serena Colafrancesco, Mario Giuseppe Bellizzi, Roberta Priori, Bruna Cerbelli, Martina Leopizzi, Carla Giordano, Antonio Greco, Paola Tirassa, Cinzia Severini, Massimo Fusconi

**Affiliations:** 1Institute of Biochemistry and Cell Biology, National Research Council (IBBC-CNR), Department of Sense Organs, Sapienza University of Rome, Viale del Policlinico 155, 00185 Rome, Italy; pamela.rosso@ibbc.cnr.it (P.R.); elena.fico@ibbc.cnr.it (E.F.); paola.tirassa@cnr.it (P.T.); 2Department of Internal Medicine and Medical Specialties, Rheumatology Unit, Sapienza University of Rome, Viale del Policlinico 155, 00185 Rome, Italy; serena.colafrancesco@uniroma1.it; 3Department of Sense Organs, Sapienza University of Rome, Viale del Policlinico 155, 00185 Rome, Italy; mariogiuseppe.bellizzi@uniroma1.it (M.G.B.); antonio.greco@uniroma1.it (A.G.); massimo.fusconi@uniroma1.it (M.F.); 4Division of Rheumatology, Department of Clinical Internal, Anaesthesiologic and Cardiovascular Sciences, Sapienza University, Viale del Policlinico 155, 00185 Rome, Italy; roberta.priori@uniroma1.it; 5Department of Medico-Surgical Sciences and Biotechnology, Sapienza University of Rome, 00185 Rome, Italy; bruna.cerbelli@uniroma1.it (B.C.); martina.leopizzi@uniroma1.it (M.L.); 6Department of Radiological, Oncological and Pathological Sciences, Sapienza University of Rome, Viale del Policlinico 155, 00185 Rome, Italy; carla.giordano@uniroma1.it

**Keywords:** substance P (SP), neurokinin receptor 1 (NK1R), minor salivary gland (MSG), primary Sjögren’s syndrome (pSS), sicca syndrome

## Abstract

Primary Sjögren’s Syndrome (pSS) is a systemic autoimmune disease that primarily attacks the lacrimal and salivary glands, resulting in impaired secretory function characterized by xerostomia and xerophthalmia. Patients with pSS have been shown to have impaired salivary gland innervation and altered circulating levels of neuropeptides thought to be a cause of decreased salivation, including substance P (SP). Using Western blot analysis and immunofluorescence studies, we examined the expression levels of SP and its preferred G protein-coupled TK Receptor 1 (NK1R) and apoptosis markers in biopsies of the minor salivary gland (MSG) from pSS patients compared with patients with idiopathic sicca syndrome. We confirmed a quantitative decrease in the amount of SP in the MSG of pSS patients and demonstrated a significant increase in NK1R levels compared with sicca subjects, indicating the involvement of SP fibers and NK1R in the impaired salivary secretion observed in pSS patients. Moreover, the increase in apoptosis (PARP-1 cleavage) in pSS patients was shown to be related to JNK phosphorylation. Since there is no satisfactory therapy for the treatment of secretory hypofunction in pSS patients, the SP pathway may be a new potential diagnostic tool or therapeutic target.

## 1. Introduction

Primary Sjögren’s Syndrome (pSS) is a systemic autoimmune disease that primarily affects the lacrimal and salivary glands, resulting in impaired secretory function characterized by xerostomia and xerophthalmia [1]. 

The characteristic autoimmune feature of pSS etiopathogenesis is based on the development of autoimmune epithelitis characterized by focal lymphocytic infiltration of the exocrine glands, often in association with acinar epithelial cell atrophy and progressive fibrosis diagnosed mainly through biopsy of the minor labial salivary glands [2].

The pathology is characterized by a broad spectrum of circulating autoantibodies, of which antinuclear antibodies (ANAs) are the most frequently detected (in approximately 70% of pSS patients) and anti-Ro/SSA and La/SSB antibodies are the most specifically detected [3]. There is convincing evidence that the innate immune response in the early phase of pathology is activated mainly by the involvement of CD4+ T cells infiltrating the salivary and lacrimal glands, whereas T cell activation and B cell accumulation occur at later stages [4]. Furthermore, activated T cells contribute to pathogenesis by releasing pro-inflammatory cytokines (e.g., TNFα, IFNs, IL-1, IL-2, and IL-6), chemokines, and increased expression of adhesion molecules, apoptosis-related factors, co-stimulatory molecules, autoantigens, and functional innate immune receptors, leading to chronic inflammatory damage to exocrine glands and progressive loss of their physiological function [5].

The prevalence of pSS is 0.15–3.3% depending on the diagnostic criteria used. Ninety-five percent of pSS patients are women, typically peri- and post-menopausal, but young post-menarche patients may also be affected [6]. Diagnostic criteria of the disease include the presence of a detectable focal lymphocytic sialadenitis (aggregates of at least 50 cells per 4 mm^2^) with a focus score ≥1, the presence of autoantibodies (i.e., anti-Ro/SSA and anti-La/SSA), and impaired salivary and ocular secretions, as defined by the American–European Consensus Group (AECG) criteria [7] and the American College of Rheumatology (ACR)/European League Against Rheumatism (EULAR) criteria [8].

Patients with pSS have been shown to have impaired innervation of the salivary glands along with changes in circulating levels of neuropeptides, which are thought to be a cause of the decreased salivation [9,10]. One possible explanation is that this autoimmune disease and/or local inflammation may cause vasoneural dysregulation and peripheral nerve injury, resulting in decreased fluid flow and acinar cell atrophy, followed by destruction of the salivary and lacrimal glands [11]. Moreover, apoptosis has emerged as a possible mechanism for damage to the salivary and lacrimal glands in pSS, leading to impairment of their secretory function [12].

Salivary secretion is controlled by the autonomic nervous system and influenced by the sensory nervous system. Activation of the parasympathetic nervous system leads to a sharp increase in salivary flow with a low protein concentration, and its function is influenced by neurons containing neuropeptides with strong sialagogue action, including substance P (SP), which is released by the nerve fibers innervating the salivary glands [13]. 

SP, a neuropeptide of 11 amino acids, was the first to be identified and is the best characterized member of the tachykinin peptide family, and its biological activity is mediated by its preferred G-protein-coupled TK receptor 1 (neurokinin receptor 1; NK1R) through the activation of phospholipase Cβ (PLCβ). Since SP and NK1R are widely distributed in the brain and sensory/autonomic nervous system, their involvement in many physiological functions can be explained [14].

In the central nervous system (CNS), SP is mainly found in neuronal cells, where it frequently co-localizes with classical transmitters and other neuropeptides and is endowed with neurotrophic and neuroprotective properties, as has been extensively demonstrated by our group and others [15,16,17,18,19].

In the peripheral nervous system (PNS), SP is found in high concentration in primary small-diameter sensory neurons (C-fibers) [20,21] and in the dorsal horns of the spinal cord, justifying its role as a sensory neurotransmitter important for pain perception [22]. In addition, SP influences smooth muscle contraction, epithelial permeability, and mediates inflammatory processes by increasing neutrophil and macrophage traffic and activating mast cells, monocytes, or lymphocytes, which release their mediators such as histamine, interleukins (IL-1, IL-2), and immunoglobulins, respectively [23].

The presence of SP nerve fibers has been demonstrated around the excretory ducts of human salivary glands, which contributes to changes in fluid and electrolyte secretion followed by a gradual increase in saliva production [24]. In salivary glands, immunoreactive SP nerve fibers have been shown to be found mainly around blood vessels and in direct contact with the acini, and their number is significantly reduced in pSS tissues [10]; however, no data are available on the presence and distribution of the associated receptor NKR1. The aim of this study was to characterize the expression of both SP and NKR1 in biopsies of the minor salivary gland (MSG) and to analyze their levels in pSS patients compared to patients with idiopathic sicca syndrome. Exploring their contribution to the pathogenesis of the disease could be useful to identify new potential diagnostic tools or therapeutic targets.

## 2. Materials and Methods

### 2.1. pSS and Sicca Patients’ Selection and Enrollment

Minor salivary gland (MSG) biopsies were obtained from patients with suspected pSS and followed up at our dedicated Sjögren Clinic at Sapienza University (Department of Clinical Internal, Anaesthesiologic, and Cardiovascular Sciences—Rheumatology Unit). The total number of patients enrolled was 30. Following the biopsy procedure, the patient was classified as pSS if they met the AECG criteria [7], while those who did not fulfill these criteria were classified as an idiopathic sicca syndrome patient. Inclusion criteria included the presence of a detectable focal lymphocytic sialadenitis (aggregates of at least 50 cells per 4 mm^2^) with a focus score ≥1, and presence of autoantibodies (i.e., anti-Ro/SSA and anti-La/SSA). Exclusion criteria included ongoing treatment with corticosteroids, hydroxychloroquine, immunosuppressants, or pilocarpine (within 3 months). Informed and written consent was obtained from patients enrolled in the study. Permission for the use of MSG samples for research purposes was obtained by our ethical committee of Sapienza University of Rome, under protocol number 4688.

### 2.2. MSG Biopsy Collection

Surgical procedures and tissue sample collection of MSG biopsies were performed by otolaryngology specialists of the Department of Sense Organs, Division of Otolaryngology, Sapienza University of Rome. A local anesthetic was injected into the lower lip, followed by a small incision in the lip mucosa. Two MSG biopsies per patient were obtained. One of the two biopsy samples was embedded in paraffin for immunohistochemistry/immunofluorescence (IHC/IF) analysis and the remaining one was immediately frozen after the surgical procedure and stored at −80 °C until it was used for Western Blot (WB) analysis.

### 2.3. Protein Extraction and WB Analysis

For protein characterization, extracts from the MSG biopsies were obtained using a lysis buffer (RIPA buffer), and the total protein concentration was measured using the BioRad protein assay (DC Protein Assay, Biorad Laboratories, CA, USA, #500-0116). WB analysis was performed using selective antibodies against substance P (SP, Santa Cruz Biotechnology, Dallas, TX, USA, sc-517213, 1:1000) and its cognate receptor Neurokinin 1 (NK1R, Santa Cruz Biotechnology, Dallas, TX, USA, sc-365091, 1:1000), cleaved PARP-1 (Santa Cruz Biotechnology, Dallas, TX, USA, sc-56196, 1:1000), and phosphorylated JNK (pJNK, cell signaling technology, Danvers, MA, USA, #9251, 1:1000). The images obtained from the WBs were analyzed using ImageJ software for Windows. All samples were normalized for protein loading using β-actin (Santa Cruz Biotechnology, Dallas, TX, USA, sc-47778, 1:10,000) as the protein loading control. The values were determined from the ratio between the arbitrary units (a.u.) derived from the protein band and the respective β-actin band and expressed as mean ± Standard Deviation (SD). 

### 2.4. Immunohistochemistry and Immunofluorescence

Immunohistochemistry (IHC) was used to characterize the infiltrates and to detect the presence of Germinal Centers (GCs). GC+ samples were defined by the presence of at least one GC (nodular aggregate segregated in T and B cells areas with positive staining for CD21L and Bcl6 on sequential sections [25]); the methods for the staining procedure were reported in our previous work [26]. 

After collection, MSGs were immediately fixed in 10% formalin and embedded in paraffin. Paraffin-embedded sections were dewaxed using 2 changes of xylol, 15 min each. After hydration in graded ethanol solutions (100%, 95%), the slides were hydrated in distilled water. The slides were then immersed in antigen retrieval solution (10 mM sodium citrate, 0.05% Tween 20, pH 6.0), using a microwave oven operated at 640 W for 15 min. After cooling, the slides were transferred to PBS 1X and incubated for 24 h at 4 °C with primary antibodies: substance P (Santa Cruz Biotechnology, sc-517213, 1:100) and e-cadherin (ECAD, ThermoFisher Scientific, Waltham, MA, USA, #53-3249-82, 1:100) in PBS 1X to perform immunofluorescence (IF). The slides were then washed in PBS 1X and incubated for 1 h at R.T. with the appropriate secondary antibody: Donkey anti-Mouse Alexa Fluor 546 (ThermoFisher Scientific, Waltham, MA, USA, A-Catalog #A10036, 1:300). The nuclei were stained with 4′,6-diamidino-2- phenylindole (DAPI, 1:10,000, SeraCare, Milford, CT, USA, KPL Dapi 1 mg 71-03-01). The slides were mounted using aqueous mounting medium. Negative controls were performed by omitting the primary antibodies (ctl neg).

All immune-reacted sections were analyzed in detail and representative images were captured using the Zeiss LSM 780 laser-scanning confocal head with a Zeiss Axio Imager Z1 microscope; LSM510 Image Examiner Software was used to process the images, which were then composed into figures using Adobe illustrator 27.4 and Photoshop CS6.

To quantify SP immunoreactivity, 5 (40× magnification) images were captured for each group. During the image acquisitions, the exposure parameters, such as gain and time, were kept constant, to avoid observing differences among experimental groups due to artifacts. The analysis of SP IF staining (%Area SP^+^) was performed using the ImageJ software (version 1.53, National Institutes of Health, Bethesda, MD, USA, https://imagej.nih.gov/ij/download.html) and the statistical significance was calculated using the t-Student test, *p* < 0.001 (***).

### 2.5. Data Analysis 

Statistical analyses were performed using GraphPad InStat3 for Windows. The data from WBs were analyzed using one-way analysis of variance (ANOVA) followed by a Tukey’s post hoc test. All results are expressed as mean ± SD, with n being the number of independent experiments. The significance level was set at *p* < 0.05 (*), *p* < 0.01 (**), and *p* < 0.001 (***).

## 3. Results

### 3.1. Lower Expression of SP and Higher Expression of NK1R in MSGs of pSS Patients Compared with Sicca Patients 

As previously reported, immunohistochemical experiments indicated a reduction in SP in pSS [10], but no data are available on quantitative protein expression of SP and its associated receptor (NK1R).

Here, we demonstrated by Western blot analysis of MSG extracts that the levels of SP and NK1R significantly differed between the two groups. The level of SP was significantly decreased in the pSS patients compared with the sicca group (** *p* < 0.01) as shown in Figure 1A. In contrast, NK1R showed the opposite trend (Figure 1B) with a significant increase in pSS patients compared to sicca patients (*** *p* < 0.001).

### 3.2. SP and ECAD Localization in MSG Sections from pSS and Sicca Patients’ Immunofluorescence

Using Immunofluorescence (IF) technique, we investigated the expression of substance P in MSG sections from pSS and sicca patients. We detected that SP expression (red) in pSS MSGs was reduced when compared to sicca MSGs, particularly in ductal epithelial cells (Figure 2A). When quantitatively analyzed, SP immunoreactivity levels appeared significantly lower in pSS patients compared to sicca individuals (*** *p* < 0.001, t = 6.302; Figure 2B). To assess the primary antibody specificity, a negative control (ctl neg) was also performed (first row, Figure 2A).

To further investigate the localization of SP in the MSGs of sicca and pSS patients, a double IF analysis for SP and E-Cadherin (ECAD), a membranous glycoprotein expressed in epithelial cells, was performed. SP localization was detected in both sicca and pSS patients, in ductal and acinar cells, and mainly in the epithelial compartment as shown by the merge images of both groups (Figure 3). 

### 3.3. Western Blot Analysis Reveals Increase in PARP-1 Cleavage and JNK Phosphorylation in MSGs of pSS Patients

Since apoptosis is involved in the pathogenesis of pSS [27], we investigated proteolysis of poly (ADP-ribose) polymerase-1 (PARP-1), an enzyme involved in DNA repair [28], and the activation of c-Jun N-terminal kinases (JNK) known to trigger the caspase pathway. To this end, we performed Western blotting analysis with an antibody that recognizes the 89 kDa fragment released by caspase-3-mediated cleavage of PARP-1 and an antibody against the phosphorylated form of JNK (pJNK).

The representative bands of cleaved PARP-1 (c-PARP-1) and pJNK WBs are shown in Figure 4A,B, respectively. They show a significant increase in both proteins in the MSGs of pSS compared with those of sicca patients (* *p* < 0.05).

## 4. Discussion

Over the past few years, the role of SP has been intensely investigated in different systems such as inflammatory bowel disease (IBD) [23], in motor and non-motor Parkinson’s disease (PD) [29], and amyloid precursor protein (APP) metabolism in an Alzheimer’s disease (AD) model [17].

SP could be also involved in the impairment of salivary secretion observed in pSS patients. Treatments for pSS-related xerostomia have been classified as symptomatic, topical, or systemic stimulants, and regenerative treatments, but there is no satisfactory therapy to treat salivary hypofunction. The result of various clinical trials is that parasympathomimetics, including pilocarpine, able to stimulate exocrine gland secretion, are the most beneficial treatments for the management of salivary dysfunction in pSS [30].

Although a subjective, transient improvement in salivary function has been consistently observed with pilocarpine therapy, the exact mechanism is unclear. It has been hypothesized that pilocarpine could improve salivary secretion through local stimulation of neuropeptidergic fibers. The results of Sato et al. [31] in healthy humans showed that oral treatment with pilocarpine triggered an increased release of neuropeptides, including SP, in both saliva and blood, suggesting that this sialagogue neuropeptide should be an important factor contributing to the stimulatory pilocarpine treatment.

To confirm this hypothesis, preclinical experiments were performed in rats that showed a comparable effect on salivary secretion when treated with SP or with parasympathomimetics. Interestingly, the use of a selective SP antagonist has been shown to reduce the increased salivary secretion elicited by pilocarpine treatment [32]. These data confirm that the stimulatory effect of pilocarpine on rat submandibular glands is not only due to activation of muscarinic receptors, but probably also to the direct stimulation of SP nerve fibers.

The contribution of the SP pathway to the pathogenesis of the disease could be useful to identify new therapeutic targets, as already suggested in studies on pSS and other forms of keratoconjunctivitis sicca [14].

Indeed, clinical trials with the administration of SP analogs (eledoisin and physalaemin) in the form of eye drops have shown significant improvement in tear secretion in these patients [33,34,35].

Additional evidence is provided by the NOD (non-obese diabetic) mouse model of SS, which is genetically predisposed to the development of autoimmune exocrinopathy and mimics the sialoadenitis of Sjögren’s syndrome [36]. 

According to these previous studies, our results also point toward SP’s involvement in pSS, showing that SP levels strongly decreased in pSS MSGs compared to sicca MSGs, when analyzed by WB. This result was confirmed by the low level of SP detected using IF in pSS ductal epithelial cells, as reported in Figure 2, as well as in both acinar and ductal cells, detected in combination with ECAD IF, in Figure 3. The role of SP in the epithelium has been also studied in other tissues like the eye, where it has been found to be protective towards diabetes-related wounds in the corneal epithelium, when interacting with its related receptor NK1R [37].

Consistent with our results, the salivary glands of these mice, together with the presence of autoantibodies, were characterized by decreased levels of β-adrenergic, muscarinic, and neuropeptide signal transduction responses and by increased levels of apoptosis. The authors proposed that apoptosis is a key factor in the development of the disease and triggers autoimmunity [38].

In agreement with this finding and other clinical evidence [27], we demonstrated an increase in cleaved PARP-1 (c-PARP-1) in the MSGs of pSS patients, confirming that excessive apoptosis may contribute to pSS salivary gland dysfunction.

As previously reported, c-Jun N-terminal kinase (JNK) signaling was shown to mediate apoptosis in human salivary gland cells in vitro [39]. We confirmed the key role of this signaling pathway in pSS patients’ MSGs, showing a significant increase in JNK activation (pJNK) associated with enhanced apoptotic processes.

The therapeutic possibility of intervening and attempting to restore or stimulate salivary gland functions associated with pSS is essential from a clinical perspective, especially to prevent the complications of decreased or absent salivation [40].

Since pilocarpine treatment is the most effective treatment for improving salivary secretion by locally stimulating neuropeptidergic fibers and, in particular, by increasing the production and release of SP, this underscores the importance of our experimental results with regard to possible drug therapy, as previously suggested [33,34,35]. 

Furthermore, because there is no satisfactory therapy for the treatment of secretory hypofunction in pSS patients, the new data on the upregulation of the NK1 receptor represent a nodal point in the possibility of using low doses of SP or synthetic selective NK1 agonists to treat pSS symptoms.

## Figures and Tables

**Figure 1 cells-12-01347-f001:**
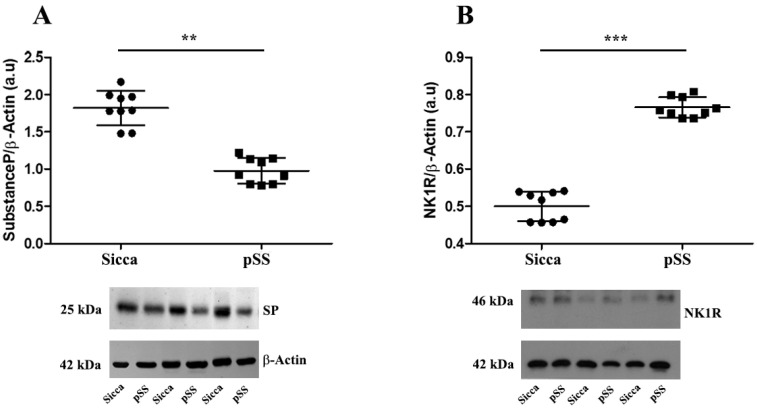
(**A**,**B**) Representative Western blot of SP (**A**) and NK1R (**B**) levels in MSGs from sicca and pSS patients. Data are expressed as mean optical density in arbitrary units (a.u.) and are given as mean ± SD of n = 9 independent experiments. Statistical significance calculated using one-way analysis of variance (ANOVA) for repeated measures followed by Tukey’s post hoc test, indicated with ** *p* < 0.01; *** *p* < 0.001. β-actin expression was used to normalize sample variability. Molecular weight markers (kDa) are shown on the left.

**Figure 2 cells-12-01347-f002:**
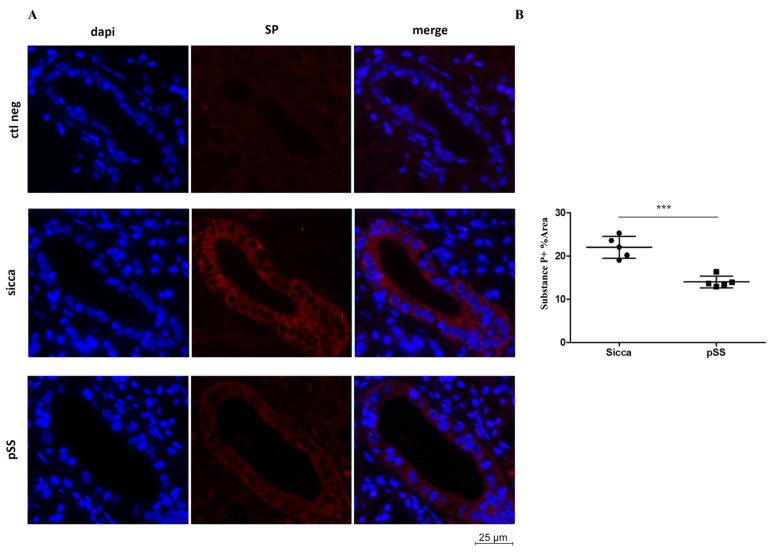
(**A**,**B**) SP expression and localization was severely affected in MSG sections from pSS and sicca patients. (**A**) Representative images of immunofluorescence staining (red) of SP in pSS and sicca patient MSG sections. (**B**) Quantitative analysis showing lower expression of SP in pSS compared to sicca patients. n = 5 independent experiments and statistical significance calculated using t-Student test. *** *p* < 0.001; t = 6.302. Scale bar 25 µm; 40× magnification. First row: negative control (ctl neg).

**Figure 3 cells-12-01347-f003:**
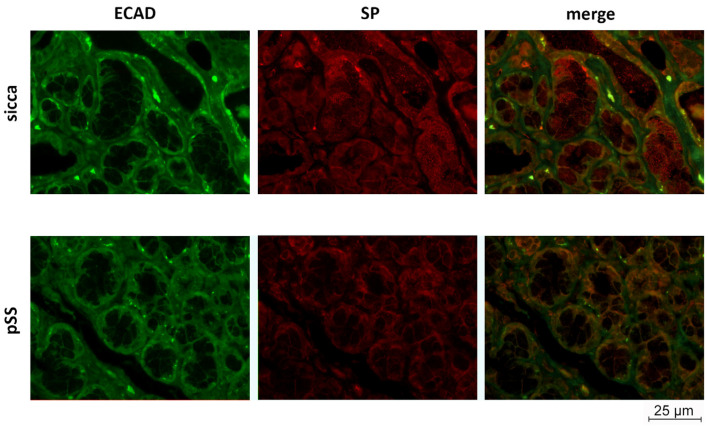
SP and ECAD expression and localization was severely affected in MSG sections from pSS and sicca patients. Representative images of double immunofluorescence staining of substance P (SP, red) and e-cadherin (ECAD, green) in pSS and sicca patients MSG sections. Scale bar 25 µm; 40× magnification.

**Figure 4 cells-12-01347-f004:**
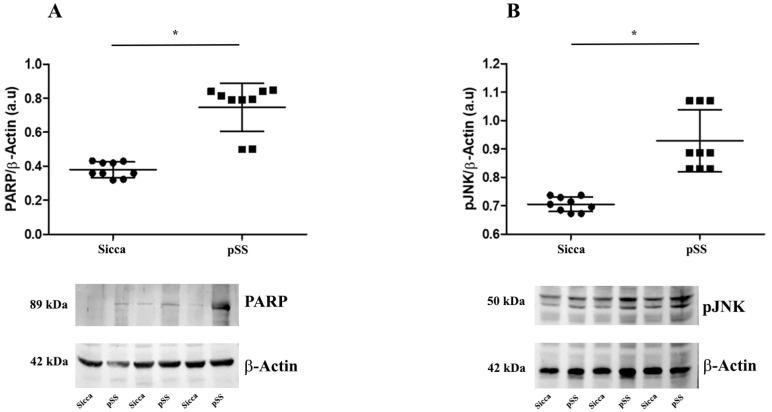
(**A**,**B**) Representative Western blot of c-PARP-1 (**A**) and pJNK (**B**) levels in MSGs from sicca and pSS patients. Data are expressed as mean optical density in arbitrary units (a.u.) and are given as mean ± SD of n = 9 independent experiments. Statistical significance was calculated using one-way analysis of variance (ANOVA) for repeated measures followed by Tukey’s post hoc test, indicated with * *p* < 0.05. β-actin expression was used to normalize sample variability. Molecular weight markers (kDa) are shown on the left.

## Data Availability

Data presented in this study are available on request from the corresponding author.

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
