# Peer review of "Involvement of Substance P (SP) and Its Related NK1 Receptor in Primary Sjögren’s Syndrome (pSS) Pathogenesis"

_cells, 2023, doi:10.3390/cells12101347_

Round 1

Reviewer 1 Report

This manuscript aims to analyse minor salivary gland biopsies from primary SS patients versus sicca patients, in order to determine if there are differences in the neuropeptide Substance P (SP). The study is interesting and well designed. However, the data presented needs some improvement in order to be of publishable quality. The following should be undertaken or addressed:

Please present the data in Figure 1A and B by displaying all data points, not just the average and SD (error bars). Likewise in Figure 3A and B.

Please include n numbers in the figure legends (e.g. Figure 1A and B) and the statistical test used. Likewise in Figure 3.

Figure 1: Legend states “Data are… given as means ± SD” but graphs only show +SD. Please correct legend. Likewise in Figure 3.

Figure 2: please include scale bars.

Please quantify the IF data presented in Figure 2. While SP certainly appears to be reduced in pSS patient biopsies compared to sicca biopsies, this must be quantified to make the claim “SP expression 171 (red) in pSS is reduced when compared to sicca MSG…”. Furthermore, in order to state “in particular in ductal epithelial cells” an epithelial marker (e.g. EpCAM or ECAD) must also be included in this figure.

In order to support the statement “In our study, the results indicate the involvement of SP fibers in the impaired salivary secretion observed in pSS patients” (line 200) and to ascertain whether the SP staining is indeed in nerve fibres or not, a nerve marker (e.g. beta-III-tubulin) should be included in the IF panel. Without this, the statement above is not correct, the data merely shows that there is less SP staining in pSS compared to sicca patients. Similarly with the statement “This result is confirmed by the low level of SP detected using IF in pSS ductal epithelial cells, as reported in Figure 2 (line 225)”. Without epithelial co-staining this statement is unsubstantiated. Moreover, are the authors suggesting that SP produced by nerve fibres, or epithelial cells, or both is reduced in pSS? The data presented suggests that SP produced by epithelial cells is reduced, but the Discussion statement starts by mentioning SP fibres.

Reviewer 2 Report

The manuscript entitled "Involvement of Substance P (SP) and its related NK1 Receptor in the primary Sjögren's Syndrome (pSS) pathogenesis" is a good work done by the authors. The manuscript need some modifications prior to acceptance.

1. The molecular basis of pathology of Primary Sjögren's Syndrome needs to be included in detail in the introduction

2. In methodology, the description on statistical comparison is to be included

3. Likewise, the antibodies used in the study with their make should be mentioned in the methodology

4. Discussion section could be elaborated by including the future perspectives of the study

Round 2

Reviewer 2 Report

No more comments